# Three-Dimensional Imaging for Multiplex Phenotypic Analysis of Pancreatic Microtumors Grown on a Minipillar Array Chip

**DOI:** 10.3390/cancers12123662

**Published:** 2020-12-07

**Authors:** Min-Suk Oh, Iftikhar Ali Khawar, Dong Woo Lee, Jong Kook Park, Hyo-Jeong Kuh

**Affiliations:** 1Department of Biomedicine & Health Sciences, Graduate School, The Catholic University of Korea, Seoul 06591, Korea; ohming1@catholic.ac.kr; 2Department of Medical Life Sciences, College of Medicine, The Catholic University of Korea, Seoul 06591, Korea; sakhawatali59@catholic.ac.kr; 3Department of Urology, Samsung Advanced Institute of Health Science and Technology (SAIHST), Samsung Medical Center, Sungkyunkwan University, Seoul 06351, Korea; 4Departments of Biomedical Engineering, Konyang University, Daejeon 35365, Korea; mems@konyang.ac.kr; 5Department of Biomedical Science, Research Institute for Bioscience & Biotechnology, Hallym University, Chuncheon 24252, Korea; jkp555@hallym.ac.kr; 6Cancer Evolution Research Center, College of Medicine, The Catholic University of Korea, Seoul 06591, Korea

**Keywords:** tumor spheroid, microtumor model, tissue optical clearing, 3D imaging, minipillar array chip, tumor invasion, extracellular matrix remodeling

## Abstract

**Simple Summary:**

Three-dimensional (3D) culture of tumor spheroids (TSs) within the extracellular matrix (ECM) recapitulates solid tumors in vivo. This microtumor model is particularly useful for multiplex phenotypic analysis, but requires tissue optical clearing (TOC) for 3D visualization. We developed a transfer-free 3D microtumor culture-to-3D visualization system using a minipillar array chip combined with the TOC method. Our method succeeded in improving immunostaining and optical transmission in each TS as well as the entire microtumor specimen. The utility of this method was demonstrated by showing phenotypic changes, such as increased levels of membrane protrusion, single-cell dissemination, and ECM remodeling, and changes in the expression of epithelial–mesenchymal transition–related proteins and drug-induced apoptosis in TSs of human pancreatic cancer cells co-cultured with cancer-associated fibroblasts and M2-type tumor-associated macrophages.

**Abstract:**

Three-dimensional (3D) culture of tumor spheroids (TSs) within the extracellular matrix (ECM) represents a microtumor model that recapitulates human solid tumors in vivo, and is useful for 3D multiplex phenotypic analysis. However, the low efficiency of 3D culture and limited 3D visualization of microtumor specimens impose technical hurdles for the evaluation of TS-based phenotypic analysis. Here, we report a 3D microtumor culture-to-3D visualization system using a minipillar array chip combined with a tissue optical clearing (TOC) method for high-content phenotypic analysis of microtumors. To prove the utility of this method, phenotypic changes in TSs of human pancreatic cancer cells were determined by co-culture with cancer-associated fibroblasts and M2-type tumor-associated macrophages. Significant improvement was achieved in immunostaining and optical transmission in each TS as well as the entire microtumor specimen, enabling optimization in image-based analysis of the morphology, structural organization, and protein expression in cancer cells and the ECM. Changes in the invasive phenotype, including cellular morphology and expression of epithelial–mesenchymal transition-related proteins and drug-induced apoptosis under stromal cell co-culture were also successfully analyzed. Overall, our study demonstrates that a minipillar array chip combined with TOC offers a novel system for 3D culture-to-3D visualization of microtumors to facilitate high-content phenotypic analysis.

## 1. Introduction

The three-dimensional (3D) culture of cancer cells provides an in vitro tumor model that best recapitulates the in vivo tumor pathophysiology [1,2]. As the most commonly used 3D culture type, tumor spheroids (TSs) retain important 3D characteristics of in vivo tumors, which are not maintained in two-dimensional (2D) culture models, such as multicellular organization and extracellular matrix (ECM) protein deposition, thereby allowing for cell–cell and cell–ECM interactions [3,4]. Cultures of TSs embedded in a suitable hydrogel biomatrix, namely a microtumor, are particularly useful for evaluating invasive cell migration out of TSs into the ECM, and concurrent ECM remodeling in drug screening and mechanistic studies [5,6,7].

The mechanical preparation of thin sections from frozen or formalin-fixed paraffin-embedded (FFPE) 3D specimens, with subsequent staining, imaging, and image reconstruction, has been considered the standard method for visualization at the cellular resolution [8,9]. However, the troublesome and time-consuming steps of these preparation procedures often cause the potential loss of sections, which may result in imprecise spatial reconstruction, leading to partial or incomplete sample analysis and incompatibility with high-throughput screening (HTS). To overcome these issues, immunocytochemical staining of the whole 3D sample with subsequent whole-mount 3D imaging can be considered as a better alternative. However, the limited diffusion of staining macromolecules in TSs and the light scattering of microtumors up to millimeter-scale specimens impose challenges in imaging and visualization of 3D cultures due to preventing the effective labeling of the inner layer of cells and intracellular proteins of interest [10,11].

Recent progress in tissue optical clearing (TOC) technology has revolutionized deep-tissue imaging in neuroscience research, enabling visualization of the 3D neuronal structures in the mouse brain [12,13]. As a non-destructive imaging method, TOC has also shown potential in microtumor imaging and for quantitative analysis of cellular events combined with immunofluorescence staining [14,15,16]. Since membrane protrusions, tumor cell dissemination, and ECM remodeling occur in 3D space [17,18], methods of concurrent 3D imaging of TSs and the surrounding ECM are critically needed for studying the interaction between cancer cells and the tumor microenvironment (TME), and for efficient screening of agents with potential anti-invasion activity [19,20]. However, to our knowledge, the application of TOC to microtumors has not been actively explored, requiring a simple and rapid protocol that should support the practical usefulness of 3D microtumors.

We previously developed a minipillar array-based chip system as a novel microtumor model in which the TSs and stromal cells were grown in a collagen matrix and co-cultured to allow for mutual interactions [11,21]. Pitch-tunable minipillar array chips have also been developed for increasing the throughput in the culture and preparation of mechanical sections of frozen or FFPE microtumors, providing significant advantages in multiplex analysis of multiple cell types under various culture conditions [22]. Here, we applied a TOC method to microtumors cultured on minipillar array chips to visualize TSs and the surrounding ECM in the entire 3D spatial context. We developed a protocol for the on-chip imaging of intact 3D microtumor specimens following in situ immunocytochemical staining using a pillar array platform, which ultimately allowed for 3D culture to 3D visualization of microtumors in a transfer-free manner. The utility of this system for high-content 3D phenotypic and molecular analyses of cancer invasion was further demonstrated using human pancreatic TSs of Capan-1 and PANC-1 cells co-cultured with tumor-promoting stromal cells such as cancer-associated fibroblasts (CAFs) or M2-type tumor-associated macrophages (TAMs). The present study suggests the great potential of our method in studies of invasion mechanisms and anti-invasive cancer drug discovery.

## 2. Results

### 2.1. Application of TOC for Microtumors Grown in a Collagen Matrix

The TOC method was adapted to improve the immunostaining and imaging of microtumors cultured on minipillar array chips. Pancreatic cancer cells (Capan-1 and PANC-1) formed multiple TSs within the collagen matrix after 6 days of culture, which was considered to be an in vitro tumor. In the minipillars, microtumor samples were subjected to tissue-hydrogel polymerization and lipid extraction, followed by immunostaining and imaging. Tissue-hydrogel polymerization and lipid extraction were carried out either in a passive manner using a vacuum desiccator and an oven or in an active manner using commercial equipment (X-CLARITY™ system, Figure 1A). Active clearing dramatically increased the speed of lipid extraction, taking less than 2 h instead of the 3 days required for passive clearing. The throughput of sample processing was further increased using a jig designed to hold six minipillar array chips in an electrophoresis chamber (Appendix A). Following tissue clearing, immunostaining of microtumors was performed by exposing the cells to primary and secondary antibodies in 96-well plates. Confocal imaging of microtumors was performed either directly on the minipillar or on a confocal dish after sample transfer (Figure 1A).

After tissue clearing, light transmittance was significantly enhanced, as shown in microtumors stained with DAPI (Figure 1B). The light transmittance of microtumors was improved from as low as 36% in uncleared samples to 94% in cleared samples, in the range of visible wavelengths from 400 to 700 nm. These results indicated that TOC was successfully applied to microtumors cultured on a minipillar array chip in a transfer-free manner throughout the procedure from 3D culture to 3D visualization.

### 2.2. Light Scattering and Limited Antibody Penetration in TSs

The feasibility of antibody staining and imaging of uncleared TSs were evaluated using Capan-1 cells. After 6 days of culture, multiple TSs formed within a microtumor with a size range of 80–90 μm in diameter. After staining with β-actin antibody and DAPI, optical sections of a single TS were acquired and 3D-rendered images were reconstructed with z-stack data (Figure 2A). A significant reduction in the fluorescence signal in the TS was observed along the *z*-axis; the fluorescence intensity of β-actin and DAPI decreased by 1.5-fold and 1.9-fold, respectively, at the 65-μm plane compared with that at the 5-μm plane.

To explain the fluorescence attenuation in the inner layer of TSs, cryosections and optical sections of uncleared samples at z = 40 μm were compared (Figure 2B). Homogeneous and strong DAPI fluorescence signal in cryosections indicated no penetration barrier for DAPI staining in microtumors. However, significant decrease in DAPI fluorescence intensity was observed in the optical sections of uncleared samples, suggesting the presence of light scattering beyond the 40-μm depth in TSs. On the other hand, fluorescence intensity of β-actin in the cryosections was significantly lower than that of DAPI supporting that not only light scattering, but also the limited penetration of staining antibodies caused the attenuation of fluorescence intensity in the inner region of the TS. These results demonstrated that even in a TS with a size less than 100 μm in diameter, optical tissue clearing was required to improve 3D imaging by resolving light scattering as well as the limited penetration of immunostaining antibodies.

### 2.3. Increased Antibody Penetration and Light Transmission in TSs by Tissue Clearing

Capan-1 microtumors were cultured on minipillars for 6 days and subjected to tissue clearing. Cytoskeleton and nuclear staining were performed using β-actin immunostaining and DAPI, optical sections of a single TS were acquired, and 3D-rendered images were reconstructed with z-stack data. In contrast to the uncleared microtumor samples (Figure 2A), the fluorescence signal was evenly detectable in the cleared TS for both β-actin and DAPI (Figure 3A). When compared to uncleared samples, the fluorescence intensity of cleared samples increased by 2.8-fold and 6.3-fold for DAPI and β-actin, respectively (Figure 3B). The signal levels of DAPI in the cleared TS were comparable with those in cryosections (Figure 2B), indicating that light scattering was eliminated by tissue clearing. The increase in the fluorescence signal of β-actin was much greater than that of DAPI, which indicated that both antibody penetration and light transmittance were significantly improved by TOC for β-actin staining. These results demonstrated that tissue clearing was successfully applied to enable the 3D visualization of specific molecular markers in a TS within microtumors using conventional immunostaining and confocal microscopy (Figure 3C).

### 2.4. Tissue Clearing Improved 3D Imaging of the Surrounding Matrix

Optical sections were obtained and 3D images were reconstructed with z-stack data following immunostaining of type I collagen and vimentin, along with nuclear DAPI counterstaining of uncleared and cleared microtumors. Collagen expression was found in ECM and between cells in TS with negligible contribution from the cell cytoplasm (Appendix A). The fluorescence signals in the uncleared microtumors were significantly attenuated beyond a 50-μm z-depth and became negligible at a 200-μm z-depth (Figure 4A). In contrast, the effective level of fluorescence signals was detected throughout the entire depth of the microtumors (z = 450 μm) following tissue clearing.

To explain the fluorescence signal attenuation in the deeper region of the microtumor collagen matrix, cryosections and optical z-sections of uncleared samples were compared for the fluorescence area and intensity of type I collagen. In cryosections, fluorescence attenuation was negligible throughout the entire depth (z = ~450 μm) of the microtumors (Figure 4B and Appendix A). In optical sections of uncleared samples; however, the Col-1 fluorescence area and intensity at a depth of 150 μm were attenuated by 60% and 63% respectively compared to cryosections. This indicated that there was no barrier in the penetration and distribution of staining antibodies in the ECM of microtumors; rather, the signal attenuation was due to limited light transmission in the deeper region of uncleared microtumors. The optical sections of cleared samples showed significant improvement in fluorescence signal detection, as shown by insignificant decrease in the fluorescence area throughout the entire depth of the microtumors (Figure 4B). Nonetheless, a 23% decrease was observed in the fluorescence intensity at a depth of 150 μm in the cleared samples, suggesting that partial light scattering remained in the deeper region of the microtumor. Overall, these results suggested that TOC was successfully adapted for 3D imaging of the ECM of microtumors (z = 450 μm) grown on minipillar array chips (Figure 4C).

### 2.5. Analysis of ECM Remodeling

The phenotypic changes associated with ECM remodeling were evaluated in PANC-1 microtumors after TOC and immunostaining of type I collagen for structural organization of the ECM. Compared with TS without invasive protrusion, increased ECM remodeling was observed in the region near the protrusions with polarized alignment of matrix fibers in the direction of invadopodia formation (Figure 5). Although the overall florescence intensity of the collagen matrix was similar, significant changes in collagen matrix topology were observed in terms of fiber area, thickness, and coherency.

### 2.6. Multiplex Analysis of Changes in Cellular Protein Expression Induced under Stromal Cell Co-Culturing

Successful multiplex analysis of cellular protein expression was demonstrated in Capan-1 microtumors following TOC and subsequent immunostaining. The expression levels of epithelial–mesenchymal transition (EMT)—related proteins such as E-cadherin, vimentin, alpha-smooth muscle actin (α-SMA), and transforming growth factor-beta 1 (TGF-β1) were evaluated in Capan-1 microtumors grown either with pancreatic stellate cells (PSCs) under bi-culture conditions or with both PSCs and TAMs under tri-culture conditions. The 3D visualization of cellular proteins was achieved in individual TSs and microtumors at all depths (Figure 6A). Changes in the protein expression levels were found under stromal cell co-culture conditions in a target-dependent manner as follows (Figure 6A). A significant reduction in E-cadherin expression was observed only under tri-culture conditions, but not under PSC bi-culture conditions, whereas significant changes in the expression levels of other proteins were observed under both co-culture conditions. The vimentin expression level increased under PSC co-culture conditions, and a further increase was noted under PSC and TAM tri-culture conditions. α-SMA levels increased under PSC co-culture conditions, but were unexpectedly suppressed to a significantly lower level than that observed under monoculture in the PSC and TAM tri-culture conditions. A similar level of increase in TGF-β1 expression was observed under both co-culture conditions, indicating no additional effect by TAMs. Drug-induced apoptosis and the effect of stromal cell co-culturing were compared in Capan-1 microtumors when exposed to 10 nM of gemcitabine or 3 nM of paclitaxel using cleaved caspase-3 expression as a marker of apoptosis. These drug concentrations induced a 30% decrease in the viability of cancer cells as determined by the calcein-AM assay (Appendix A). The level of cleaved caspase-3 expression decreased by 19% following treatment with gemcitabine and by 28% following treatment with paclitaxel under both stromal cell co-culture conditions, indicating that PSC co-culture contributed to drug resistance and no resistance was further conferred by the presence of TAMs in the culture (Figure 6B).

### 2.7. Analysis of Invasive Morphological Phenotypes in TSs

The phenotypic changes associated with invasive transition were evaluated in PANC-1 microtumors after TOC and immunostaining of vimentin for membrane protrusions (Figure 7A). No significant difference in the size of TSs was observed between TSs monoculture and TSs co-cultured with PSCs. When microtumors were cultured under PSC co-culture conditions, the number of protrusions per TS (1.4-fold), their length (1.4-fold), and the number of single cells disseminated from the TS (1.45-fold) was increased (Figure 7B). This indicated that PANC-1 microtumors assumed invasive phenotypic changes under PSC co-culture conditions, probably via the soluble factors released by PSCs. Optimized analysis of membrane protrusions and cell dissemination was performed in the entire microtumor, as its 3D images were successfully reconstructed after TOC; that is, quantitative analysis of membrane protrusions growing in all directions was only possible in 3D-rendered images of whole microtumor samples.

## 3. Discussion

In vitro 3D tumor models, such as TSs, are known to retain the important 3D characteristics of in vivo tumors and their TME, and to recapitulate the molecular signature and behavior of cancer cells [24]. However, critical limitations exist in the immunostaining and visualization of such 3D models due to two major inherent obstacles: (1) the limited penetration of macromolecules such as staining antibodies and (2) light scattering. Mechanical sectioning is an available alternative [8,9]; however, the laborious and skill-demanding procedure is unfavorable, and 3D rendering from 2D sections is unreliable [25,26]. In this study, we successfully applied TOC to microtumor samples cultured on minipillar array chips. The method presented in this paper represents an efficient 3D culture to 3D imaging platform, enabling quantitative analysis of cellular morphology and behavior, and cell–cell and cell–matrix interactions can be studied using even fragile samples, such as TSs and microtumors in a transfer-free manner.

The major issues in the high-content imaging of microtumor cultures are related to technical difficulties and low throughput during TOC application to microtumor samples or TSs [27]. Owing to the sub-milliliter size and fragile nature of microtumor samples or TSs, special fixing and multistep handling processes are required, especially when the 3D cultures are grown by conventional methods, such as hanging drops or ultra-low attachment plates [28]. Our platform provides significant advantages in these aspects. The inter-pillar distance in an array chip was tuned to be compatible with each well of 96-well plates so that the microtumors were cultured in conjunction with 96-well plates. TOC and immunostaining were performed in 96-well plates without requiring sample harvest or transfer. Higher throughput in lipid clearing was achieved by using an active clearing device and a jig holding six chips per batch, which allowed for up to 48 microtumor samples to be cultured per 96-well plate and cleared in a batch (Appendix A). When microtumor samples are transferred to and visualized in microfluidic-type chamber slides, long-term storage and archiving of samples can be subsequently performed. The hydrogel-hybridization and clearing steps used in TOC selectively remove lipids from the tissues while preserving the tissue architecture, native antigens, and soluble proteins with negligible loss. When brain tissues were cleared with the TOC method, complete lipid removal with minimum protein loss (~8%) was reported [12,29]. Overall, our method represents a novel system for 3D culture-to-3D imaging of microtumors in vitro, by which sample integrity can be preserved and increased throughput can be achieved.

Penetration of macromolecules in the 3D culture of cancer cells is limited because cellular lipids act as a barrier to molecular transport within the ECM and between cells [30,31]. We performed cryosectioning of the microtumor after immunostaining with β-actin to check the penetration depth of the antibody in the intact microtumor. This resulted in partial surface staining, as exemplified by β-actin staining in the cryosectioned TS (Figure 2B). Differential penetration of nanoparticles depending on size has been studied in cancer cells, TSs, and in vivo tumors [32]. Gold nanoparticles with molecular diameters of 2 and 6 nm showed unlimited penetration and reached the core of the tumor tissue, whereas nanoparticles with a diameter of 15 nm were detected only near the surface of the tumor. The molecular diameter of DAPI (molecular weight 277 Da) in aqueous solution is expected to be between 1 and 2 nm, as predicted from its binding properties as a minor-groove binder, whereas the size of antibodies such as monoclonal IgG1 (148 kDa) against β-actin approaches 10–20 nm [33,34]. Therefore, the difference in staining depth between these two agents can be attributed, at least in part, to the size difference.

Although antibody penetration in TSs was improved after TOC (Figure 3), immunostaining in a cleared microtumor sample remains a challenge for two main reasons. First, diffusion of the antibody takes a long time; it usually takes several days for antibody molecules to distribute and bind throughout the cleared 3D samples. Novel immunostaining methods following TOC have been suggested to expedite the transport of antibody molecules into cleared 3D samples. Various approaches, including membrane disruption [16], stochastic electrotransport [35], and enhancement of antibody permeability and binding affinity [36], have been tested for achieving rapid, uniform, and system-wide delivery of various antibody molecules into various types of 3D samples. Second, phalloidin, a bicyclic peptide toxin that exclusively binds to F-actin, is widely used for the study of F-actin–mediated cellular processes and morphology [37]. Unfortunately, phalloidin staining is not compatible with TOC; hence, in the present study, β-actin and vimentin staining was used instead for identification of cell morphology and membrane protrusions, respectively. A search for novel alternatives to fluorophore-conjugated phalloidin for F-actin staining is therefore highly warranted.

In this study, we successfully immunostained and imaged individual TSs as well as whole microtumors generated from Capan-1 and PANC-1 cells (Figure 3 and Figure 4 and Appendix A). Our previous report showed that five different pancreatic cancer cell lines can be categorized according to their basal epithelial/mesenchymal phenotype [21]. Depending on their basal phenotype, the pancreatic cancer cell lines showed different patterns of EMT level, ECM remodeling, and plasticity of cell migration, classified as individual or collective migration [20]. Therefore, Capan-1 cells, which have an epithelial phenotype, were utilized for EMT and drug response analysis due to their dramatic phenotypic changes induced under stromal cells co-culture conditions (Figure 6). The PANC-1 cell line, which is known to exhibit a mesenchymal phenotype, was appropriate for the analysis of individual cell migration out of the TS, and was hence useful in the evaluation of protrusion formation, ECM remodeling, and cell dissemination (Figure 5 and Figure 7).

It is evident that reciprocal interactions between cancer cells and stromal cells promote EMT, cell invasion, ECM remodeling, and drug resistance [38,39]. We provided an efficient method to culture multiple TSs within microtumors along with stromal cells, such as CAFs and TAMs, to mimic the in vivo TME. The usefulness of our method is highlighted in multiplex platforms where various culture conditions and drug exposure can be evaluated within a single batch of experiments, which can facilitate multiple comparisons and correlation tests. In addition, the 96-well plate-based layout provides easy integration with a commercial liquid-handling robot and automated reader, which will eventually contribute to improving the throughput of the screening system.

## 4. Materials and Methods

### 4.1. Cell Culture and Reagents

The human pancreatic cancer cell lines PANC-1 and Capan-1 were purchased from American Type Culture Collection (Manassas, VA, USA) and Korean Cell Line Bank (KCLB; Seoul, Korea), respectively. Human pancreatic stellate cells (PSCs) and a human monocytic cell line (THP-1) were obtained from ScienCell (HPaSteC; 3830, Carlsbad, CA, USA) and KCLB, respectively. Capan-1 and THP-1 cells were maintained in RPMI-1640 medium (Gibco, Grand Island, NY, USA), whereas PANC-1 and PSCs were maintained in high-glucose Dulbecco’s modified Eagle medium (HyClone, Logan, UT, USA). All media were supplemented with 100 μg/mL streptomycin, 100 units/mL penicillin, 250 ng/mL amphotericin B, and 10% fetal bovine serum (Welgene, Daegu, Korea). THP-1 cells were induced to differentiate into M2 macrophages (TAMs) using 50 ng/mL phorbol 12-myristate 13-acetate (Sigma-Aldrich, St. Louis, MO, USA) and 20 ng/mL of interleukin-4 (PeproTech, Cranbury, NJ, USA) as confirmed by CD-206 expression. Cell culture was maintained in a humidified atmosphere (5% CO_2_/95% air) at 37 °C.

### 4.2. Minipillar Array Chips and a Jig for Spheroid Culture and Tissue Clearing

The minipillar array chips were custom-made for the culture and optical clearing of microtumors by MBD Co., Ltd. (Su-won, Korea) [21]. Minipillar array chips specially designed for TOC consisted of a reusable pillar holder (43 × 15 × 6 mm) and eight disposable minipillars (^∅^3.2 × 13 mm, 2 × 4 array) (Appendix A). The minipillars were made of polystyrene by injection molding, as previously reported [22]. The pillar holders were prepared from an acrylonitrile butadiene styrene copolymer using a computer numerical control machine. Sterilization was performed using a 70% ethanol wash followed by ultraviolet irradiation (Ultraviolet Crosslinkers; UVP CX-2000, Analytik Jena AG, Jena, Germany) as previously described [21,22]. For the active clearing process, a polycarbonate jig (55 × 17 × 75 mm) was used to hold up to six minipillar chips (Appendix A, MBD).

### 4.3. Culture of Microtumor on Minipillar Array Chips

All cells were suspended in 2 mg/mL collagen I solution (rat tail tendon type I collagen, Corning, NY, USA). For Capan-1 and PANC-1 cells, 2 μL of a cell-collagen suspension was loaded onto the tips of the minipillars at 800 and 400 cells/pillar, respectively. PSCs and TAMs were plated at a density of 1600 cells each in 3 μL of a cell-collagen mixture in 96-well plates. After 5 min of gelation in a cell culture incubator, the 96-well plates were filled with culture media, and the cell-laden minipillars were incubated tip-side down to grow microtumors after 6 days of culture. When co-culturing, culture media were selected according to the growth media of the cancer cell lines and were replenished every 2 days.

### 4.4. TOC

Tissue clearing was performed according to the CLARITY protocol described by Chung et al. [23,29]. In brief, tissue-hydrogel polymerization was performed in 96-well plates that were pre-filled with a paraformaldehyde-containing hydrogel solution (C13103, Aligned Genetics, Inc., An-yang, Korea). For passive tissue clearing, de-gassing was performed in a vacuum desiccator for 10 min, followed by incubation at 37 °C for 3 h. Lipid extraction was performed using an electrophoretic tissue clearing solution (C13001, Aligned Genetics, Inc.) for 3 days. For active tissue clearing, hydrogel polymerization and lipid clearing processes were carried out using the C20001 and C30001 X-CLARITY™ systems, respectively (Aligned Genetics, Inc.). The light transmittance of the microtumors was measured using a NanoDrop™ 2000 spectrophotometer (Thermo Fisher Scientific, Waltham, MA, USA).

### 4.5. Immunofluorescence Staining

Immunofluorescence staining was performed directly in 96-well plates following TOC. After blocking non-specific binding using 10% normal goat serum for 6 h, samples were incubated with the following primary antibodies at 4 °C for 18 h: β-actin (1:300, sc-47778, Santa Cruz Biotechnology, Dallas, TX, USA), type I collagen (1:800, ab34710, Abcam, Cambridge, UK), vimentin (1:600, ab92547, Abcam), E-cadherin (1:300, 3195S, Cell Signaling Technology, Danvers, MD, USA), transforming growth factor beta-1 (TGF-β1; 1:100, ab92486, Abcam), α-smooth muscle actin (α-SMA; 1:300, ab5694, Abcam), cleaved-caspase-3 (1:300, 9661, Cell Signaling Technology), and CD-206 (1:100, ab125028, Abcam). Alexa Fluor 594 (A27016) or Alexa Fluor 488 (A11034)-conjugated secondary antibodies were used (1:300; Thermo Fisher Scientific). DAPI was used for nuclear counterstaining (1:500, D9564, Sigma-Aldrich).

### 4.6. Image Acquisition and Analysis

Stained samples were observed under a confocal microscope (LSM 800 W/Airyscan, Carl Zeiss, Oberkochen, Germany) using X-CLARITY™ Mounting Solution (C13101, Aligned Genetics, Inc.). Fluorescence intensity was determined using the ZEN software (Carl Zeiss, San Francisco, CA, USA). Optical section images were merged to produce z-stack images or were reconstructed into 3D images (Imaris v8.1, Bitplane, Belfast, UK). Data processing and analysis were done using either whole microtumors or three to four representative TSs selected from approximately fifty TSs formed in a microtumors. Three independent experiments were performed in triplicate. The area, thickness [40], and coherency [41] of matrix fibers were analyzed using ImageJ software (National Institutes of Health, Bethesda, MD, USA). The apparent diameter (D) of TSs was calculated using the equation D = 2 × (area/π)^1/2^, assuming a spherical shape of TSs, in which the area was measured using ImageJ. The number and length of invasive protrusion analysis were performed on the fiber structure grown from the TSs using Filament Tracer (Imaris). Based on the individual cell size of 10 to 20 μm, cells less than 40 μm in diameter were counted as single cells or loose aggregates of single cells and larger than 40 μm were considered TSs. The regions of interest for image analysis were selected randomly on the designated sections, unless otherwise indicated.

### 4.7. Statistical Analysis

All data are expressed as the mean ± standard deviation of three independent experiments or measurements. Student’s t-test and analysis of variance (ANOVA), followed by Duncan’s post-hoc test, were used to test the statistical significance using Microsoft Excel 2010 and SPSS version 24 (SPSS Inc., Chicago, IL, USA). *p* values ≤ 0.05 were considered statistically significant.

## 5. Conclusions

We demonstrated that TOC was successfully applied to microtumors cultured using a minipillar array chip. Immunostaining and light transmission were significantly improved, enabling high-content image analysis of cellular and extracellular proteins. The 3D visualization and multiplex image analysis were successfully performed for quantitative phenotypic analysis including invasive protrusions, ECM remodeling, and EMT markers in an in vitro 3D tumor model. This platform represents a novel method of multiplex 3D imaging and phenotypic analysis of microtumors in vitro. This method should be useful in the discovery of molecular targets and new treatment strategies targeting cancer invasion.

## Figures and Tables

**Figure 1 cancers-12-03662-f001:**
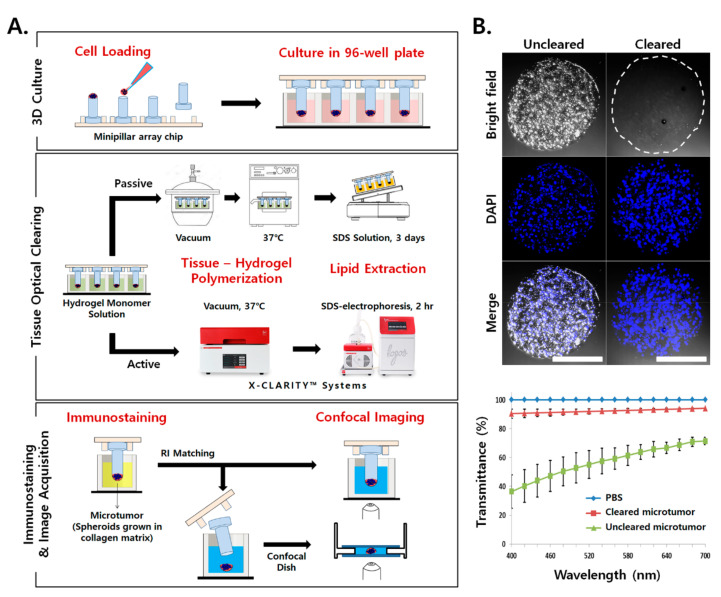
Three dimensional (3D) culture and tissue optical clearing (TOC) of microtumors grown on a minipillar array chip. (**A**) Cancer cells suspended in collagen solution were loaded onto minipillars and cultured to grow microtumors in a 96-well plate. Tissue clearing was achieved by polymerization of a tissue-hydrogel followed by sodium dodecyl sulfate–based lipid extraction using the CLARITY method [23]. A commercial X-CLARITY™ system was used for the active process. Confocal imaging of the microtumor in a 96-well plate or on a confocal dish after sample transfer was performed following immunostaining and refractive index matching. (**B**) Improved light transmittance by TOC was confirmed in microtumors stained with DAPI. Data represent the mean ± SD of three independent experiments. Scale bars, 1 mm.

**Figure 2 cancers-12-03662-f002:**
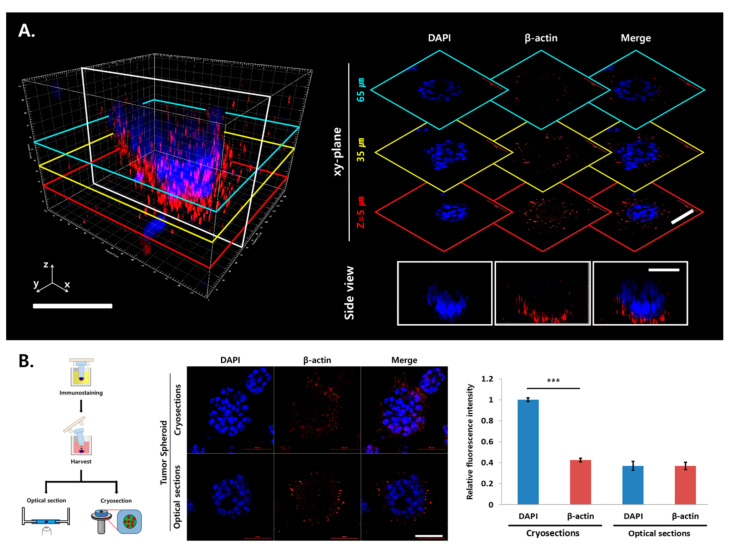
Limited penetration of immunolabeling antibodies in Capan-1 tumor spheroids (TSs). (**A**) Three-dimensional (3D)-rendered confocal images of a single Capan-1 TS following DAPI and β-actin staining. Optical sections were obtained at 5-μm increments along the *z*-axis for 3D reconstruction. (**B**) Fluorescence intensity of DAPI and β-actin was compared between optical sections and cryosections at z = 40 μm. Cryosections were prepared at 10 µm thickness. Whole specimen of microtumor was stained with β-actin antibody and DAPI and then subjected to optical section or cryosections. Data represent the mean ± SD of three independent experiments. *** *p* < 0.001. Scale bars: 50 μm.

**Figure 3 cancers-12-03662-f003:**
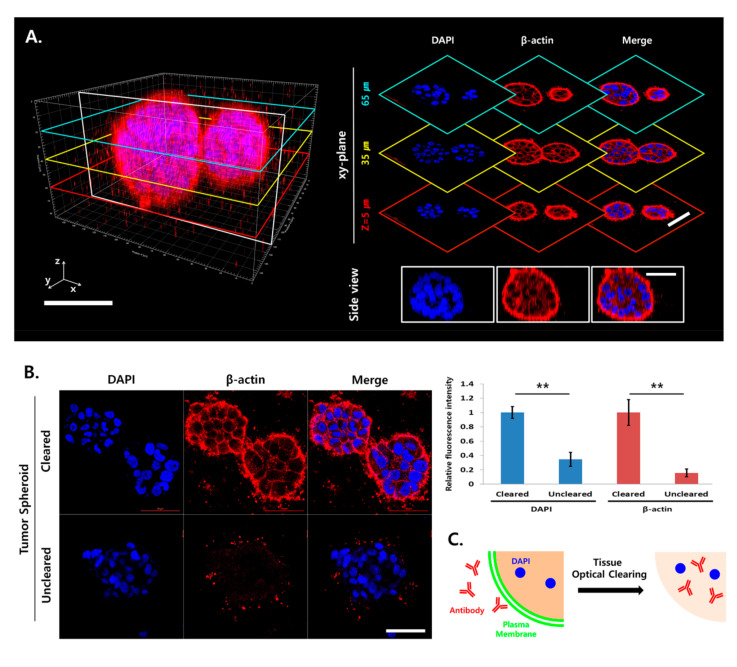
Improved penetration of immunolabeling antibodies and light transmission in Capan-1 tumor spheroids (TSs) after tissue optical clearing (TOC). (**A**) Three-dimensional (3D) confocal images of a single Capan-1 TS stained with β-actin (red) and DAPI (blue). Optical sections were obtained at 5-μm increments along the *z*-axis for 3D reconstruction. (**B**) Increased fluorescence intensity of β-actin (red) and DAPI (blue) in optical section images of TS shown in cleared samples. (**C**) Schematic illustration of improved penetration of immunolabeling antibodies in TS following lipid extraction during TOC. Data represent the mean ± SD of three independent experiments. ** *p* < 0.01. Scale bars: 50 μm.

**Figure 4 cancers-12-03662-f004:**
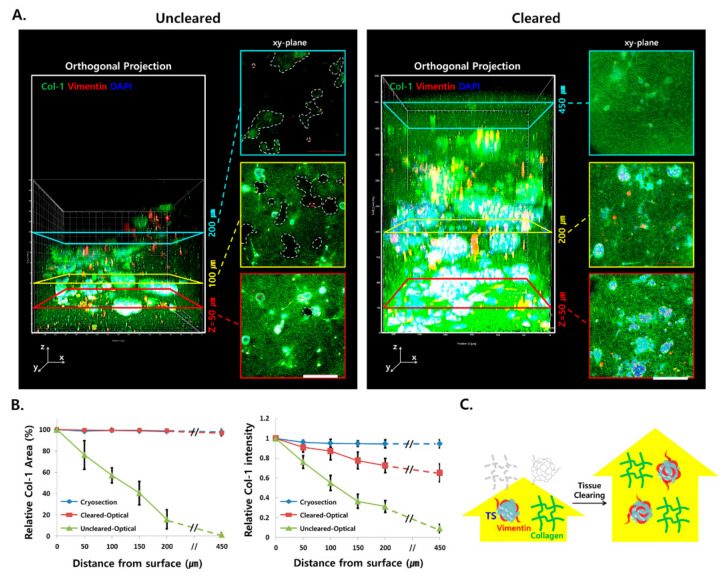
Increased imaging depth in Capan-1 microtumor following tissue optical clearing (TOC). (**A**) Comparison of 3D-rendered optical images of a microtumor with or without TOC. Optical sections were acquired at 5-μm intervals along the *z*-axis for 3D reconstruction. (**B**) Fluorescence area and intensity of collagen over 450-μm distance in cryosections and optical sections with or without TOC. The fluorescence area and intensity were shown relative to those at the z = 0 μm section. (**C**) Schematic illustration of improved light transmittance by TOC in microtumors stained with type I collagen (green), vimentin (red), and DAPI (blue). Data represent the mean ± SD of three independent experiments. Scale bars: 200 μm. Col-1; type I collagen.

**Figure 5 cancers-12-03662-f005:**
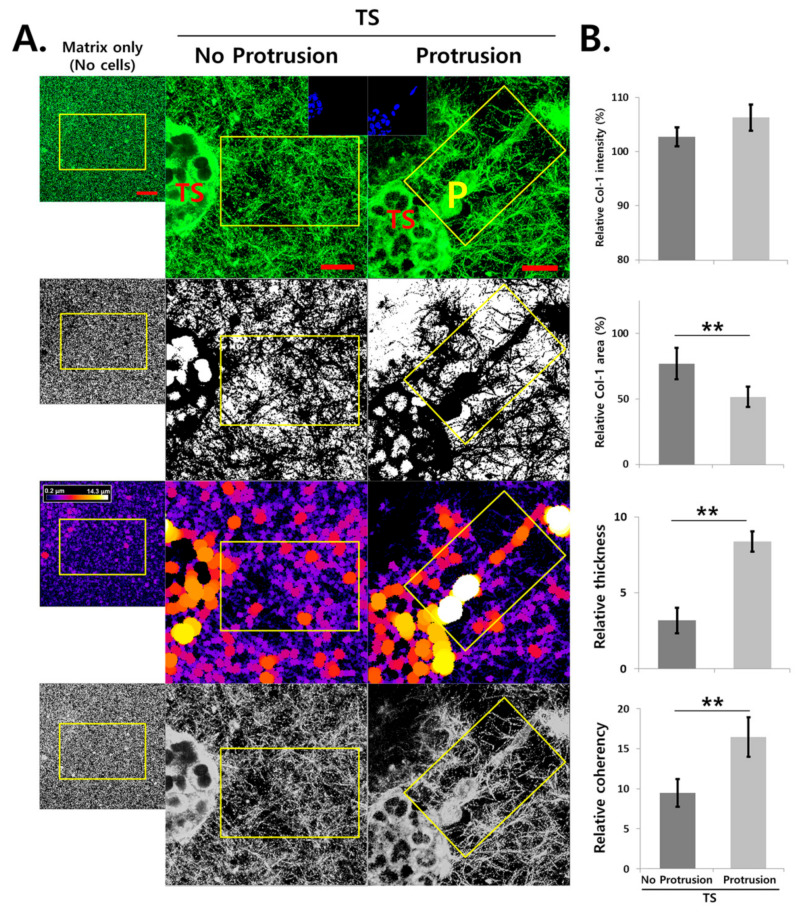
Analysis of extracellular matrix remodeling. (**A**) Representative optical sections and processed images showing fiber distribution and organization of collagen matrix in the areas close and distant to protrusions. (**B**) Quantitative analysis was done in rectangular (80 μm × 50 μm) regions-of-interest (ROI, yellow box) around or distal to membrane protrusions. Intensity and distribution area of Col-1 and matrix fiber topology were shown relative to those determined in matrix only samples cultured without cells. PANC-1 tumor spheroids were cultured for 6 days. Image analysis was performed by ImageJ. Scale bars: 20 μm. ** *p* < 0.01. TS; tumor spheroid, P; protrusion, Col-1; type I collagen.

**Figure 6 cancers-12-03662-f006:**
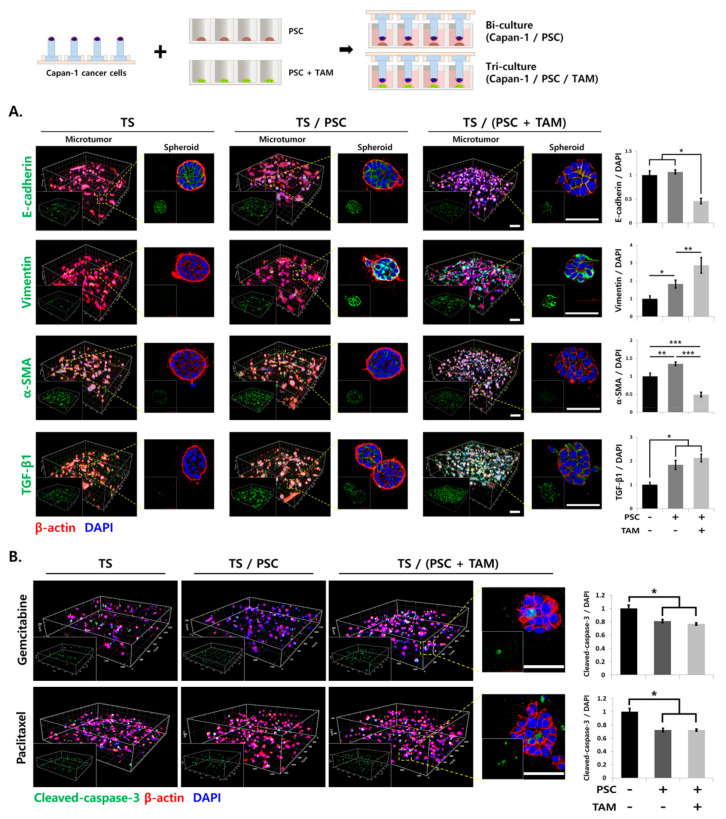
Effect of stromal cell co-culture on epithelial mesenchymal transition (EMT) and drug sensitivity of Capan-1 TSs. (**A**) Changes in the expression levels of EMT-related proteins in Capan-1 tumor spheroids (TS) under bi-culture (PSCs) or tri-culture (PSCs + tumor-associated macrophages (TAMs)) conditions. (**B**) Effect of stromal cell co-culture on drug-induced apoptosis in Capan-1 TSs. Confocal sections of microtumors were acquired at 3-μm intervals and 3D images of a microtumor and an individual TS were reconstructed. Schematic illustration of the co-culture condition is shown at the top. The EMT-related markers (E-cadherin, vimentin, α-SMA, TGF-β1) and apoptosis marker (cleaved-caspase-3) are shown in green, β-actin is shown in red, and DAPI is in blue. Data represent the mean ± SD of three independent experiments. Scale bars: 50 μm. * *p* < 0.05, ** *p* < 0.01, *** *p* < 0.001. TS: tumor spheroid; PSC, pancreatic stellate cell; TAM, tumor-associated macrophage; α-SMA, alpha-smooth muscle actin; TGF-β1, transforming growth factor-beta1.

**Figure 7 cancers-12-03662-f007:**
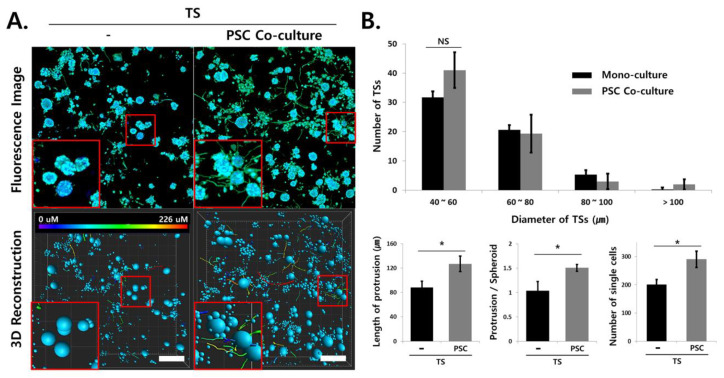
Analysis of single-cell dissemination and plasma membrane protrusion. (**A**) Invasive protrusions from tumor spheroids (TSs) and single-cell dissemination were analyzed in three dimensional-reconstructed images of the whole microtumor. Insert: representative images of individual TSs. Membrane protrusion was traced by immunostaining of vimentin. Protrusions with varying lengths are shown using spectral colors. (**B**) Image analysis was performed by ImageJ (number and size of spheroids) and Imaris (protrusion and single cell dissemination). PANC-1 TSs were cultured with or without pancreatic stellate cell for 6 days. Scale bars: 100 μm. * *p* < 0.05. NS; Non-significant, TS; tumor spheroid, PSC; pancreatic stellate cell.

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
