# Peer review of "Three-Dimensional Imaging for Multiplex Phenotypic Analysis of Pancreatic Microtumors Grown on a Minipillar Array Chip"

_cancers, 2020, doi:10.3390/cancers12123662_

Round 1

Reviewer 1 Report

In this revision, Min-Suk Oh and et al included some major changes and the writing, derivations, and methods are now presented much more clearly, and as such, the manuscript has improved significantly. Below, I presented a couple of new criticisms and suggestions.

  1. The author stated that the “relative fluorescence area was calculated by normalizing fluorescence area to the stained area of first image section”. However, in figure 5, we can observe fluorescence signal inside the cell cytoplasm. A single normalization would induce errors in evaluating the collagen area. However, the authors provided no details if this was considered or excluded from the analysis. If so, how was the signal from the cells excluded?

  1. Figure 5 lacks information in the legend on what the yellow box highlighted areas are.

  1. The authors still lack information on how the quantitative data analysis were done. Are all the TSs in each microtumors included in the analysis or only representative TSs were chosen? If so, how many TSs from each microtumors were included?

  1. I appreciated that the authors took time to further explain the reasons for different intensity level between beta-actin and DAPI. However, it still didn’t address my concern.

The authors claimed the intensity decay of DAPI between cryosections and optical sections is due to light scattering (2-fold). However, the fluorescence intensity of beta-actin stain between cryosections and optical sections are on the similar level (figure 2). How come the light scattering did not affect the fluorescence intensity of beta-actin?

Author Response

[Response to Reviewer 1 (Minor revision)]

Here, we enclosed our response to each point raised by reviewer. We made due changes to address all the comments from reviewer. We believe that our revised manuscript has been substantially improved. We now hope that is in the acceptable form to your prestigious journal.

Point 1. The author stated that the “relative fluorescence area was calculated by normalizing fluorescence area to the stained area of first image section”. However, in figure 5, we can observe fluorescence signal inside the cell cytoplasm. A single normalization would induce errors in evaluating the collagen area. However, the authors provided no details if this was considered or excluded from the analysis. If so, how was the signal from the cells excluded?

We appreciated reviewer’s comments that helped us correct errors in data analysis and presentation as follows.

Figure 5: Graphs were modified after analysis was re-done to the ROI’s defined outside of TSs. Legend was changed accordingly.

Legend of Figure 5: Analysis of extracellular matrix remodeling. (A) Representative optical sections and processed images showing fiber distribution and organization of collagen matrix in the areas close and distant to protrusions. (B) Quantitative analysis was done in rectangular (80 μm × 50 μm) regions-of-interest (ROI, yellow box) around or distal to membrane protrusions. Intensity and distribution area of Col-1 and matrix fiber topology were shown relative to those determined in matrix only samples cultured without cells. PANC-1 tumor spheroids were cultured for 6 days. Image analysis was performed by ImageJ. Scale bars: 20 μm. **p < 0.01. TS; tumor spheroid, P; protrusion, Col-1; type I collagen.

Point 2. Figure 5 lacks information in the legend on what the yellow box highlighted areas are

We explained this point in the answers to point 1.

Point 3. The authors still lack information on how the quantitative data analysis were done. Are all the TSs in each microtumors included in the analysis or only representative TSs were chosen? If so, how many TSs from each microtumors were included?

We made changes in Methods (page 14, line 424) to provide details for quantitative data analysis as follows.

Methods (page 14, line 424): … Data processing and analysis were done using either whole microtumors or three to four representative TSs selected from approximately fifty TSs formed in a microtumors. Three independent experiments were performed in triplicate.

Point 4. I appreciated that the authors took time to further explain the reasons for different intensity level between beta-actin and DAPI. However, it still didn’t address my concern.

The authors claimed the intensity decay of DAPI between cryosections and optical sections is due to light scattering (2-fold). However, the fluorescence intensity of beta-actin stain between cryosections and optical sections are on the similar level (figure 2). How come the light scattering did not affect the fluorescence intensity of beta-actin?

We revised the whole paragraph and mentioned limited antibody penetration along with light scattering as a cause for similar level of β-actin fluorescence intensity between cryo- and optical sections in the Result (page 4, line 135) as follows.

Result (page 4, line 135): To explain the fluorescence attenuation in the inner layer of TSs, cryosections and optical sections of uncleared samples at z = 40 μm were compared (Figure 2B). Homogeneous and strong DAPI fluorescence signal in cryosections indicated no penetration barrier for DAPI staining in microtumors. However, significant decrease in DAPI fluorescence intensity was observed in the optical sections of uncleared samples, indicating the presence of light scattering beyond the 40-μm depth. On the other hand, fluorescence intensity of β-actin in the cryosections in TSs was significantly lower than that of DAPI indicating that not only light scattering but also limited penetration of staining antibodies caused the attenuation of fluorescence intensity in the inner region of the TS. These results suggest that even in a TS with a size less than 100 μm in diameter, optical tissue clearing is required to improve 3D imaging by resolving light scattering as well as the limited penetration of immunostaining antibodies.

Additional changes: We modified Result 2.4 (page 6, line 177) to explain the results more clearly as follows.

Result 2.4 (page 6, line 177): 2.4. Tissue clearing improved 3D imaging of the surrounding matrix

Optical sections were obtained and 3D images were reconstructed with z-stack data following immunostaining of type I collagen and vimentin, along with nuclear DAPI counterstaining of uncleared and cleared microtumors. Collagen expression was found in ECM and between cells in TS with negligible contribution from the cell cytoplasm (Supplementary Figure S3). The fluorescence signals in the uncleared microtumors were significantly attenuated beyond a 50-μm z-depth and became negligible at a 200-μm z-depth (Figure 4A). In contrast, the effective level of fluorescence signals was detected throughout the entire depth of the microtumors (z = 450 μm) following tissue clearing.

To explain the fluorescence signal attenuation in the deeper region of the microtumor collagen matrix, cryosections and optical z-sections of uncleared samples were compared for the fluorescence area and intensity of type I collagen. In cryosections fluorescence attenuation was negligible throughout the entire depth (z = ~450 μm) of the microtumors (Figure 4B, Figure S2). In optical sections of uncleared samples, however, the Col-1 fluorescence area and intensity at a depth of 150 μm were attenuated by 60% and 63% respectively compared to cryosections. This indicated that there was no barrier in the penetration and distribution of staining antibodies in the ECM of microtumors; rather, the signal attenuation was due to limited light transmission in the deeper region of uncleared microtumors. The optical sections of cleared samples showed significant improvement in fluorescence signal detection, as shown by insignificant decrease in the fluorescence area throughout the entire depth of the microtumors (Figure 4B). Nonetheless, a 23% decrease was observed in the fluorescence intensity at a depth of 150 μm in the cleared samples, suggesting that partial light scattering remained in the deeper region of the microtumor. Overall, these results suggested that TOC was successfully adapted for 3D imaging of the ECM of microtumors (z = 450 μm) grown on minipillar array chips (Figure 4C).

Thank you for your consideration of publication of our manuscript.

Hyo-JeongKuh, Ph.D.

College of Medicine, The Catholic University of Korea

222, Banpo-daero, Seocho-gu,

Seoul 06591, Republic of Korea

Tel) 82-2-2258-7511

Email) [email protected]

Reviewer 2 Report

Accept

Author Response

Accepted

This manuscript is a resubmission of an earlier submission. The following is a list of the peer review reports and author responses from that submission.

Round 1

Reviewer 1 Report

In this manuscript, the authors assessed the potential of using tissue optical clearing (TOC) on three-dimensional cultured prostate tumor spheroids (TS). By combining with their minipillar array chip, the authors were able to achieve a transfer-free 3D microtumor culture to 3D visualization pipeline. The authors proved that the TOC was able to greatly increase the imaging depth and improve the penetration of antibodies for better staining. With the set-up, the authors were able to evaluate the invasive phenotype of TS when co-cultured with pancreatic stellate cells and observed changes in cellular protein expressions induced under stromal cell co-culturing or drug treatment. The ability to visualize the entire 3D culture without sectioning is important. In fact, using CLARITY TOC method on 3D cultures has been reported and utilized by a number of groups (Chen et al. 2013, 2016, 2019).

The manuscript is well written and organized. However, it can benefit of significant improvement given the authors address the following:

  1. Please specify the reason why different pancreatic cancer cell lines were used in different experiments. The authors used PANC-1 in section 2.5 to evaluate the invasive phenotypes when co-cultured with PSCs and used CAPAN-1 for all other studies. What is rational?

The authors claimed that there were phenotypic changes associated with invasive transition (section 2.5) and there were protein expression changes induced by stromal cell co-culture or drug exposure (section 2.6). To support this claim, the authors need to include more than one cell line to rule out the possibility that the response to the cell co-culture and drug treatment was cell-type specific.

  1. The authors lack a lot of details in the image analysis. One critical information the authors did not provide is the number of spheroids that was included in the study. Did the author imaged multiple gel cultured independently or all the analysis was done with multiple TS in one gel?

  1. Same as point 2, the authors evaluated “fluorescence area” (Line 173) of type I collagen in both the uncleared and cleared sample. However, in the figures, the y axis was labelled Relative Col-1 area. Were these two the same? How were the “fluorescence area”/Relative Col-1 area calculated?

  1. Same as point 2, the authors didn’t specify how the fiber density, thickness and coherency (Line 208) was evaluated/calculated.

  1. The authors claimed that the microtumors demonstrated significant increase of the number of protrusions per TS, their length and the number of single cells disseminated from the TS when co-cultured with PCSs. Again, the authors lacked critical information in terms of their analysis. It is known that the size of TS can also have an effect on the number of protrusions, their length and the number of disseminated cells. Did the authors compare the size of the TS w/o co-culture? Does the trend still hold true if the authors only include the TS that were of similar diameter? How did the authors quantify the number of disseminated? On the inset of figure 5a, the TS without PSC co-cultured showed clearly more single cells.

Minor:

  1. The authors should change the range of y axis for the “-actin (relative intensity)” in figure 2B so that more details can be visualized. Also, do the authors know why the intensity of -actin between the optical sections and cryosections did not change? Shouldn’t light scattering affect both DAPI and -actin?

  1. Please include captions for the green membrane in figure 3C.

  1. Please also include the original 3D stacks for figure 5A.

Reviewer 2 Report

The authors reported a 3D cell culture and clearing system for high content imaging microtumor in a chip. The idea of this paper is interesting however the experimental design is not proper. The paper will need at least major revision to be accepted for publication. The comments are as follows.

  1. It is not clear what exactly is developed by the authors reading the first several sections. It seems that they just used a commercially available system. They need to clearly specify what is their original contribution.
  2. FFPE slides has been widely used for analyzing biomarkers in tumor spheroid, which is also very efficient. What is the advantage of their platform then? Just for 3D images?
  3. Why the antibody cannot stain the cryosection? There should be no penetration issue for the 10-um slide. This data does not make sense.
  4. Will the tissue clearing cause the loss of soluble biomarkers?
  5. The section of TAM coculture part is not clear. It seems that this part has no scientific discovery other than some fancy pictures.
  6. The in vitro model of co-cultured cell lines is too far away from in vivo realistic. The authors should try to match their discovery to in vivo results or at least tumor organoid.